# Limited Acclimation in Leaf Morphology and Anatomy to Experimental Drought in Temperate Forest Species

**DOI:** 10.3390/biology11081186

**Published:** 2022-08-07

**Authors:** Attaullah Khan, Fangyuan Shen, Lixue Yang, Wei Xing, Brent Clothier

**Affiliations:** 1Key Laboratory of Sustainable Forest Ecosystem Management-Ministry of Education, School of Forestry, Northeast Forestry University, Harbin 150040, China; 2Sustainable Production, New Zealand Institute for Plant & Food Research Limited, Tennent Drive, Palmerston North 4474, New Zealand

**Keywords:** *Larix gmelinii*, *Pinus koraiensis*, *Fraxinus mandshurica*, *Tilia amurensis*, drought tolerant, leaf hydraulic traits, biomass allocation, mesophyll

## Abstract

**Simple Summary:**

Climate change shown to have a significant impact on the forest ecosystem due to increased and more frequent occurrence of extreme drought. However, in order to successfully adjust to the xeric environments, plants can usually adopt a variety of adaptation strategies. Here, we investigated the morpho-anatomical traits and biomass allocation patterns as acclimation mechanisms in drought conditions. We found that the interrelation between leaf morphological and anatomical traits were equally affected by drought conditions across all species. This suggests that there is no convincing evidence to classify taxa based on drought resistance vs. drought tolerance. However, based on the biomass allocation pattern, we found that *P. koraiensis* and *F. mandshurica* had the higher RMF and total PB, but lower LFM, suggesting higher drought tolerance than those of the other species. Therefore, our dataset revealed some easily measurable traits, such as LMF, RMF, and PB, which demonstrated the seedling’s ability to cope with drought and which could be utilized to choose drought-tolerant species for reforestation in the temperate forest.

**Abstract:**

Drought is a critical and increasingly common abiotic factor that has impacts on plant structures and functioning and is a challenge for the successful management of forest ecosystems. Here, we test the shifts in leaf morpho-anatomical or hydraulic traits and plant growth above ground caused by drought. A factorial experiment was conducted with two gymnosperms (*Larix gmelinii* and *Pinus koraiensis*) and two angiosperms (*Fraxinus mandshurica* and *Tilia amurensis*), tree species grown under three varying drought intensities in NE China. Considering all the species studied, the plant height (PH), root collar diameter (RCD), and plant biomass (PB) were significantly decreased by drought. The leaf thickness (LT) increased, while the leaf area (LA) decreased with drought intensity. In the gymnosperms, the mesophyll thickness (MT) increased, and the resin duct decreased, while in the angiosperms the palisade mesophyll thickness (PMT), the spongy mesophyll thickness (SMT), and the abaxial (ABE) and adaxial epidermis (ADE) thickness were increased by drought. The correlation analysis revealed that *P. koraiensis* and *F. mandshurica* had the higher RMF and total plant biomass, but the least LMF, suggesting drought tolerance. In contrast, the *L. gmelinii* had the least RMF and higher LMF, suggesting vulnerability to drought. Similarly, *T. amurensis* had the higher leaf size, which increased the evaporative demand and depleted the soil water quickly relative to the other species. The interrelation among the morpho-anatomical leaf traits was equally affected by drought across all the studied species, suggesting that there is no clear evidence to differentiate the taxa based on drought resistance vs. drought tolerance. Thus, we have identified some easily measurable traits (i.e., LMF, RMF, and PB) which evidenced the seedling’s ability to cope with drought and which therefore could be used as proxies in the selection of drought tolerant species for reforestation in the temperate forest.

## 1. Introduction

Drought has been considered as an important factor in the decrease in plant performance in a changing climate [1,2]. Drought has had a negative impact on the forest ecosystem in northeast China; they have strongly influenced tree growth and survival and species distribution, as well as forest ecosystem function, structure, and productivity [3]. The above-ground plant organs of the leaves and forest ecosystems have acquired significant attention with their responses to the changing environment [4,5,6]. Little attention, however, has been focused on the variations in leaf traits of the individual trees in response to climate change [7,8]. Therefore, leaf traits are considered to be an important factor in the response to climate change for the entire ecosystem [8,9].

The relative growth of root collar diameter or stem diameter at breast height under drought compared to the growth under optimal conditions has been used to define the drought resistance of the tree species [10,11]. The drought resistance of the species depends not only on extrinsic factors (i.e., competition and habitat) but also on intrinsic factors (i.e., phenotype and genotype) [12]. The studies on intrinsic factors have investigated the molecular, physiological, and morphological traits of the species’ resistance mechanisms [13,14]. Amongst numerous leaf traits, those linked with light capture, CO_2_, and water trade-offs have acquired great attention, reflecting the critical significance of these processes in the functioning of the biosphere [15]. Yet, the leaves are highly plastic in response to their growing condition, and they vary greatly in physiology, morphology, and anatomy across species [16,17]. The larger variability in LT and LA under drought alters the leaf photosynthetic capacity (A_max_), as previously reported [18,19]. Leaves developed under drought do have a low rate of expansion, and therefore, the cells are more tightly packed and smaller, with a lower fraction of air space [19]. Small leaves under drought are more advantageous relative to large leaves as smaller leaves feature a high boundary layer conductance that lowers the leaf surface temperature by preventing heat accumulation [20,21].

The drought tolerance of trees may vary with drought severity, plant functional types, and geographic position. To date, numerous studies have been carried out to evaluate the impacts of drought on the morpho-anatomical and physiological responses among plant functional types [22,23]. It has been shown that drought considerably modifies the biomass allocation pattern and can impede the dry mass production [24,25], whereas it can enhance the root: shoot ratio [22]. Greater investment in the production of roots at the cost of shoot growth is one way of optimizing water uptake [26]; this investment strategy can increase leaf gas exchange for greater seedling productivity, which can result in greater survival in a drought-prone region [27]. For example, Poorter et al. [28] reported that drought stress generally decreased LMF and SMF, while it increased RMF. However, there is still a lack of knowledge about the alterations in biomass allocation, including leaf morpho-anatomical traits, along the environmental gradients.

When challenged with drought, most of the tree species decrease stomatal conductance to consume less water per C assimilated, thereby enhancing water use efficiency [23,29]. In water-limited regimes, plants prioritize nutrient and C investment in water absorption in order to be less susceptible to drought [27]. In response to drought stress, plants can control transpiration water loss by reducing the leaf area expansion to prevent dehydration of leaf tissues [22,30]. Under drought, the leaf metabolic responses of woody species are accompanied by stomata conductance, water potentials, and nutrient contents [31]. Some species can maintain a positive photosynthetic activity via osmotic adjustment even at low leaf and xylem water potentials [32]. Under extreme drought, a reduction in photosynthetic activity is attributed to both stomatal and non-stomatal limitations [33].

In this study, we selected the four important afforestation and timber production tree species in northeast (NE) China—*Larix gmelinii* and *Pinus koraiensis* are conifers while *Fraxinus mandshurica* and *Tilia amurensis* are compound-leaved tree species; they are all the dominant trees in temperate forests (i.e., both natural forests and plantations). These four species have contrasting drought tolerances and show different growth patterns in the area. Moreover, all the species occur in humid environments, and there is a possibility of experiencing severe drought during the growing season, which can cause catastrophic hydraulic failure or even whole-tree mortality in the future [3,34]. The experiment was conducted under controlled conditions to be able to compare species under the same stress conditions, which is difficult to realize in the field. We studied multiple functional traits, such as growth and its components, biomass allocation patterns (i.e., LMF, SMF, and RMF), and leaf morpho-anatomical and physiological traits, which allow us to tease apart which traits are the strongest drivers of drought tolerance. Here, we focused on the response of leaf physiological and morpho-anatomical traits and above-ground growth characteristics under drought, with the main objective being to test whether they are indeed the most important for drought survival and species distribution. Specifically, we proposed the following hypotheses: (1) the seedlings of the four temperate tree species differ strongly in their response to drought; (2) specific functional traits such as biomass allocation pattern, but not the leaf morpho-anatomical traits, will be good predictors for drought survival under drought.

## 2. Materials and Methods

### 2.1. Research Site

The experimental site was a plant nursery located at Jiansanjiang in Heilongjiang, China (47°15′21.0′′ N~132°37′35.0′′ E). The study area has a continental temperature monsoon climate with a mean annual temperate of 1–2 °C and a maximum temperature ranging from 20–24 °C. The experimental site receives an average rainfall of 550–660 mm with a maximum rainfall from June-August. The growing season ranges from 110–135 days. The soil is classified as chernozemic, and the fundamental soil properties are: a pH (H_2_O) of 5.57–6.28, total nitrogen (TN) of 1.11–1.46 g kg^−1^, total carbon (TC) of 73.9–104.5 g kg^−1^ and total phosphorus of 872.3–990.5 mg kg^−1^, and a high-water storage capacity.

### 2.2. Experimental Design

The tree species were *Larix gmelinii* (Rupr.) Rupr. (Dahurian larch; Lar); *Pinus koraiensis* Siebold & Zucc. (Korean pine; Pin); *Fraxinus mandshurica* Rupr. (Manchurian ash; Fra); and *Tilia amurensis* Rupr. (Amur lime; Til). Seedlings of the local genotype (per species) were purchased from the Baolongdian Forest Farm, Wuchang City, Heilongjiang Province. These are the most important tree species of the temperate region of NE China (Guo et al. 2008). These species represent different plant functional types: the Korean pine and Larch are conifers (gymnosperms), while the Ash and Lime are broad-leaved species (angiosperms).

One-year-old seedlings were transplanted with soil adhering to their roots into the plots of 1 m^2^ in May 2018. Fifty seedlings each were planted into nine plots per species. To prevent the entry of water and roots into adjacent plots, each plot was enclosed by an 80 cm-deep ditch lined with a plastic sheet. After one month of growth, all the plots were installed with a Closed Loop Irrigation^®^ system in the middle of each plot (CS3500 Model No. ACC-CON-WD64, Meridian, ID, USA 83642) to control the soil moisture regimes (i.e., severe drought, SD, ≤5%; moderate drought, MD, 6–10%; and well-watered, WW, 16–20%). The plots comprised a rainout shelter to accommodate the seedlings. The shelter was made-up of a metal frame of dimensions 20 m long by 8 m wide by 4 m high with a roof pitch of 30°. The plastic sheaths were used as a roll-up system for the roof top curtains. When there was no rain, all sides remained open to ensure uninterrupted airflow and to minimize a build-up of the ambient temperature and humidity. Whenever rain was sensed by the rain sensor installed on the shelter, the curtains were deployed. Thus, a two-factorial experiment with four species and three irrigation frequencies was established.

### 2.3. Seedling Growth and Biomass Above and Below Ground

Plant height (PH) and root collar diameter (RCD) per plot were measured every month after re-planting in the nursery using 15 randomly selected trees but excluding the trees at the exterior. Root collar diameter was determined by measuring the orthogonal diameter with digital calipers (MeasumaX IP54, Peterborough, ON, Canada). For plant height determination, the stem was pulled straight to the tallest apical bud using measuring tape.

For biomass measurements, ten randomly selected seedlings per plot were selected, excluding the trees from the exterior, and were harvested in late September 2018 and separated into root, stem, and leaves, then oven dried at 60 °C for 72 h and weighed (±0.0001 g). Plant biomass (PB; g), root–shoot ratio (root: shoot), leaf mass fraction (LMF), stem mass fraction (SMF), and root mass fraction (RMF) were calculated [35]. All the root samples were carefully collected with the hand shovel down to 60 cm.

### 2.4. Leaf Morphological and Stomatal Traits

The leaves were harvested at the end of the growing season, in September 2018, from three randomly selected trees situated in the middle of each plot (n = 9). For the gymnosperms (*L. gmelinii*, *P. koraiensis*), fifty fully developed sun-lit needles were collected from trees at the outer boundary next to the trench. For the angiosperms (*F. mandshurica*, *T. amurensis*), 5 fully developed leaves per individual on the current year shoots were harvested, placed in bags, transported to the laboratory, and stored at 2 °C. Next, the leaf samples were imaged using a scanner (600 dpi; Epson-Expression 10000XL with transparency unit, Epson, Japan). Leaf areas (LA; cm^2^) were measured with the program Motic Image Advanced v. 3.2, software (Motic Crop., Zhejiang, China). Leaf thickness (LT; µm) was calculated from multiple 8 µm-thick cross-sections with the software Motic Image Advanced v. 3.2, (see ‘Anatomy section’).

For anatomical analysis of the stomata, five branches per individual were cut, wrapped in the black plastic bags with wet filter paper in them and transported to the laboratory. Next, a total of 10 fully developed needles or leaves were excised under water and saturated overnight in distilled water. The next day, the trichome on the abaxial leaf surfaces was removed using adhesive tape, and a coat of clear nail polish was applied close to the midrib in the angiosperms, while in the gymnosperms a layer of clear nail polish was directly applied to the needle’s surfaces. The dry nail polish was then peeled off using adhesive tape and placed on a glass slide and analyzed with a compound microscope. It was assumed that the stomata would remained closed as the leaves were maintained under dark and well-watered conditions until preparation. The stomata were counted over defined areas to determine the stomata pore length (SL, µm) per individual and were measured directly on a total of up to 150 stomata using the software Motic Images Advanced v. 3.2 (see Section 2.5) [36], while the stomatal density (SD, no. mm^−2^) was calculated as the number of stomata per mm^2^.

### 2.5. Leaf Anatomical Traits 

For leaf anatomy, 10 developed and sun-lit needles or leaves from the upper canopy were taken and directly fixed in Formalin-Aceto-Alcohol solution (FAA; 5 mL 37% methanol, glacial acetic acid, and 90 mL 50% ethanol). They were immediately put in the ice box and transported to the lab and stored in a refrigerator at 4 °C. In the laboratory, the leaves and main vein of about 10.0 mm-wide pieces from the middle of the leaves per individual were taken and then dehydrated in 70, 85, 95, and 100% ethanol and colored with safranin (2%) and fast green (1%), respectively. In the gymnosperms, the slices were made in the middle of the needles, while in the angiosperms the leaves close to the mid rib were used. All the specimens were embedded in paraffin, and multiple 8 µm-thick slices were cut by microtome (KD-202, KEDEE, Jinhua, China) [37]. The images per individual were subsequently taken via compound microscope (40–1000×; Olympus Corporation, BX-51, Tokyo, Japan). The gymnosperms’ anatomical leaf traits were measured, and these included mesophyll thickness (MT; µm), a combined epi-hypodermis thickness (ETH; µm), and the resin duct (RD; µm). In the angiosperms, spongy mesophyll thickness (SMT, µm), palisade mesophyll thickness (PMT, µm), and adaxial epidermis thickness (ADE, µm) and abaxial epidermis thickness (ABE, µm) were measured. The hydraulic leaf traits that were measured were vascular bundle diameter (VBD; µm) and xylem conduit diameter (XCD; µm). The software Motic Images Advanced v. 3.2, (Motic Corp., Hangzhou, Zhejiang, China) was used for measuring the anatomical traits.

### 2.6. Stem Water Potential

Stem water potential (Ψ_stem_) was measured for 192 samples (4 species × 4 seedlings × 4 samples × 3 treatments) at the end of August 2018. Ψ_stem_ was measured with the Scholander-type pressure chamber (PMS Instrument Company, Albany, Oregon, USA) at predawn (Ψ_PD_; 4:00 am) and at midday (Ψ_MD_;12:00–14:00) on current twigs immediately upon transportation to the lab. The stem samples were collected from three central trees per plot (n = 9), because of their long pedicel (ca. 2 cm), to permit the usage of the pressure chamber. The excised stems were placed in a sealed bag for equilibrating the water potential in the samples. The bag was again put into a black bag with ice, and this method also helped in lowering the variations that occurred due to the time of collection of the samples. Stem water potential was measured within two hours in the laboratory using a magnifying lens.

### 2.7. Soil Analysis

Soil samples were collected using a soil corer in late September 2018. All the samples were placed in a plastic bag and brought back to the laboratory. Soil samples in the laboratory were air dried, and the root samples were removed through a 1 mm-size sieve mesh. Soil pH was measured using a suspension of soil water 1:2.5 (*w*/*v*) with a pH meter (MT-5000, Shanghai, China). For soil total carbon (TC) and soil total nitrogen (TN) determination, the soil was passed through 0.15 sieve and measured with an analyzer (Vario Macro Cube, Elementar Co, Langenselbold, Germany). Soil total P was determined using a flow injection analyzer (AA3 Seal Co., Ludwigshafen, Germany).

### 2.8. Statistical Analysis

A two-way ANOVA (Analysis of Variance) was performed to determine the effect of species and drought, and their interactions, on leaf functional traits and above-ground growth traits. The post-hoc Tukey’s honest significant differences (HSD) test was applied to compare treatment effects, while for normality the Shapiro–Wilk (*p* > 0.05) test was used. We then found the mean and standard error (SE) of the leaf morphological traits of leaf area (LA) and leaf thickness (LT) and the gymnosperms’ (i.e., MT, RD, and EHT) and angiosperms’ leaf anatomical traits (i.e., SMT, PMT, ADE, and ABE) and hydraulic traits (VBD, CD, SD, and SL). Redundancy analysis (RDA) was used to identify the environmental parameters, such as soil pH, TN, TC, and TP, which predict the variations in leaf functional and above-ground growth traits. An RDA analysis was carried out in R using the vegan package [38]. Pearson’s correlation analyses were performed to compare the correlation within and between leaf functional traits and the above-ground parameters. Pearson’s correlation analysis was performed with the ‘corrplot’ package in R [39]. All statistical analysis was performed in the R. software, v.3.6.1 [40]. Sigma Plot v.12.5 (Systat software Inc., San Jose, CA, USA) was used for creating bar charts.

## 3. Results

### 3.1. Seedling’s Growth Above and Below Ground

The plant traits above ground were significantly altered by severe drought (Figure 1; Table 1). Plant biomass was significantly decreased (Figure 1A; Table 1; *p* ˂ 0.01) by drought and remained stable under moderate drought at the time of harvest. The root–shoot ratio increased significantly with drought, based on a significant decrease in biomass allocation to the leaves and a significant decrease in allocation to the roots (Figure 1B,E). The biomass allocation to the stem was stable in all the studied species (Figure 1E; Table 1). We can also report that the plant height and root collar diameter were significantly decreased by severe drought and remained stable under moderate drought (Figure 1C,D). The drought resistance of the root collar diameter was lower than the plant height (data not shown).

The four species differed strongly in plant biomass and their relative allocations (Figure 1A,E). For example, *F. mandshurica* possessed the greatest total plant biomass and RMF and LMF, while it had the least SMF. In contrast, *P. koraiensis* possessed the least plant biomass and the greatest SMF, while *L. gmelinii* possessed the greatest LMF and the least RMF (Figure 1E).

### 3.2. Leaf Morphology 

The leaf morphological traits were all significantly altered by drought (Table 1). The leaf area and thickness were significantly decreased (*p* ˂ 0.05) by severe drought and remained stable under moderate drought (Figure 2A,B). A larger average increase (11.6%) in LT was observed in *P. koraiensis*, while the least average increase (7.6%) in LT was observed in *T. amurensis*; the largest average decrease (26.6%) in LA was observed in *T. amurensis* and the least average (16.3%) in LA was observed in *P. koraiensis.*

Similarly, the stomatal traits (i.e., stomatal pore length and density) were significantly modified by severe drought and remained unaffected under moderate drought (Figure 2). The SL was significantly decreased, while the SD was significantly increased by severe drought and remained stable under moderate drought (Figure 2C,D). The greatest average decrease (14.8%) in SL was observed in *F. mandshurica*, while the least average decrease (9.1%) in SL was observed in *P. koraiensis* (Figure 2C). In contrast, the greatest average decrease (16.5%) in SD was observed in *L. gmelinii* while the least average decrease (6.6%) in SD was observed in *T. amurensis* (Figure 2D).

All four species differed strongly in most of the studied leaf morphological and stomatal traits (Figure 2). For example, *P. koraiensis* possessed the greatest LT, while *T. amurensis* possessed the greatest LA (Figure 2A,B). In contrast, *F. mandshurica* possessed the greatest SL while *L. gmelinii* possessed the greatest SD (Figure 2C,D).

### 3.3. Leaf Hydraulic and Anatomical Traits and Stem Water Potentials

All the hydraulic leaf traits were significantly (*p* ˂ 0.05) modified by drought across all the species. Specifically, the vascular bundle (VBD) and conduit diameter (CD) were significantly decreased (*p* ˂ 0.05; Figure 3; Table 1) by severe drought and remained unaffected under moderate drought across all the species. The leaf anatomical traits also differed significantly for all the species. In the gymnosperms (*L. gmelinii*, *P. koraiensis*), the epi- and hypodermis (EHT) and mesophyll thickness (MT) were significantly increased (*p* ˂ 0.05, Table 1 and Table 2), while the resin duct (RD) was significantly decreased (Table 1; Figure 3) by severe drought. The ABE, ADE, PMT, and SMT were significantly increased in the angiosperms (*F. mandshurica*, *T. amurensis*) by severe drought (Table 2), while the ratios of spongy to palisade mesophyll thickness and abaxial to adaxial epidermis remained stable with drought (data not shown). 

The four species differed strongly in most of the anatomical leaf traits (Figure 3). For example, in the gymnosperms the *P. koraiensis* had the greatest resin duct and VBD diameter compared to *L. gmelinii* (Table 2). In the angiosperms, the *F. mandshurica* had the greatest mesophyll thickness, while the *T. amurensis* had the thinnest spongy and palisade mesophyll. All the species varied strongly in their hydraulic leaf traits (Figure 3). For example, *T. amurensis* possessed the larger XCD, while the *L. gmelinii* needles possessed the narrowest conduits (Figure 3B).

The stem water potentials were significantly altered by severe drought across all the species (Figure 4). The predawn (Ψ_PD_) and midday (Ψ_MD_) stem water potentials were significantly decreased by severe drought as compared to the well-watered and the moderate drought (Figure 4A,B). The Ψ_MD_ was significantly lower (more negative) than the Ψ_PD_ across all the studied species. The least (more negative) stem water potential was observed in *P. koraiensis*, followed by *L. gmelinii*, *T. amurensis*, and *F. mandshurica* (Figure 4B).

### 3.4. Interrelations of Leaf Functional Traits with Environmental Factors

A redundancy analysis (RDA) illustrated the interrelation of the leaf functional traits and the above-ground parameters to the soil-related environmental parameters (Figure 5). The first and second axis of RDA accounted for 59.9 and 3.3% of the total variations, respectively. The PB, PH, LMF, LB, RMF, SL, and XCD were strongly positively correlated to the soil-related environmental parameters (i.e., soil pH, TN, TP and TC), while LT, SD, SMF, Ψ_PD_, and Ψ_MD_ were significantly negatively correlated to the soil-related environmental parameters. However, VBD and RCD were weakly correlated to the environmental factors. Based on this RDA, up to 64.4% of the total variations were explained by environmental factors (Figure 5).

### 3.5. Correlations of Leaf Functional Traits

The trait interrelations within the leaf morpho-anatomical traits across all the studied species, i.e., *L. gmelinii* (Figure 6A), *P. Koraiensis* (Figure 6B), *F. mandshurica* (Figure 6C), and *T. amurensis* (Figure 6D) were equally affected by severe drought, while the interrelations between the leaf functional traits and the above-ground characteristics were more consistent in *P. koraiensis* and *F. mandshurica* (Figure 6B,C). LT was found to be significantly positively correlated to SD, while LA was significantly positively correlated to XCD, VBD, and SL across all the studied species (Figure 6A–D).

While LT was significantly positively correlated to RMF across *P. koraiensis*, *F. mandshurica*, and *T. amurensis* (Figure 6A–C), LT was significantly negatively correlated to PB in *L. gmelinii* and *P. koraiensis* (Figure 6A,B). LA was significantly positively correlated to LMF in *P. koraiensis*, *F. mandshurica*, and *T. amurensis*, while it was significantly negatively correlated to *P. koraiensis* and *F. mandshurica* (Figure 6B,C).

## 4. Discussion

### 4.1. Effects of Drought on Above-Ground Growth Traits

It has been widely reported that drought affects plant supply with C [41], thereby hindering plant growth [42,43]. However, the impact of drought on plant growth is difficult to compare across species because of the complex variation in drought severity, time, and duration [1,11]. In our study, we observed that the resistance of root collar diameter was lower than that of plant height, corroborating the findings of Kono et al. [10] and Bushal et al. [11]; they reported that root collar diameter was a more sensitive indicator of drought resistance than plant height. The biomass allocation pattern was significantly modified by drought, and we found a significant increase in root mass fraction and a decrease in leaf mass fraction (Figure 1E). Such a response has been linked to drought tolerance [44] but not in all cases [45]. However, our results are congruent with the meta-analysis of a large number of experimental datasets; this indicates that the dry mass allocation response to drought is strongly influenced by drought severity [28]. We observed that plants grown under severe drought often increase their investments into the root system at the expense of leaf or shoot dry mass [46,47]. Contrary to our results, some previous studies reported that drought significantly enhanced the LMF of *Setaria virdis* by up to 30% on average [48], while Hamann et al. [49] and Li et al. [23] reported a lack of allocation to roots in drought-treated herbaceous perennial species. In our study, the low LMF and higher RMF under severe drought could be linked to improved water uptake and possibly to the later shutting down of photosynthesis [42,50]. Species-specific differences in biomass allocation patterns are thought to underlie the variation in drought tolerance across species [51]. For example, a high RMF or low LMF can reduce the demand for water or for growth resources in general [45]. Furthermore, drought tolerance can be achieved by increasing the below-ground biomass allocation or by decreasing the above-ground biomass allocation and decreasing the shoot evaporative demand [45,52]. In our study, *F. mandshurica* possessed the highest RMF, while the *P. koraiensis* possessed the least LMF, compared to the rest of the species, and they also had higher dry mass, which has been linked to drought tolerance. Our findings are that the interrelations between leaf traits and above-ground growth characteristics are more consistent in *F. mandshurica* and *P. koraiensis*. These showed a similar resistant pattern with respect to plant height, root collar diameter, and biomass allocations (pattern), while the interrelations between leaf traits and above-ground growth characteristics across *L. gmelinii* and *T. amurensis* are not strong, suggesting that these species have the least vitality and are more vulnerable to drought. Our results of the greatest RMF and the least LMF by drought being associated with increased resistance to drought are confirmed by the provenance studies, where highly drought-resistant origins show reduced above-ground growth rates [53,54]. In addition, *L. gmelinii*, which exhibited the lowest performance under drought, had the least below-ground mass fraction plus a higher above-ground biomass fraction, which together result in a water-inefficient allocation pattern. Our findings are in agreement with the well-known knowledge that the investment of dry mass in the root is increased to enhance water uptake capacity under drought [26].

### 4.2. Effects of Drought on LT, LA, and Stomatal Traits 

Understanding how plant traits respond to the variability in environmental condition is important for predicting plant responses to climate change. It is not surprising that our results show that drought conditions have significantly increased LT and decreased LA across all the studied species (Figure 2A,B). In our study, a higher LT and a lower leaf size under severe drought suggest that the increased LT and decreased leaf size might have decreased the surface area for evaporative demand [55]. Our results are confirmed by the previous study of Bhusal et al. [56], who reported that the leaf size was significantly decreased while the LT was increased by drought in two cultivars of Fuji and Hongro, while Ren et al. [57] reported that the leaf size was significantly increased in the arid and semi-arid region with increasing precipitation. In our study, the interrelations between leaf morphological traits are consistent across all the studied species. This suggests that morphological traits might be more influenced by phylogeny and thus less flexible in their response to drought than physiological traits [58], particularly at the seedling stage. The higher total leaf area in *T. amurensis* under drought may offer some disadvantages as large size implies higher total leaf area in absolute terms [58]. This may contribute to a higher evaporative demand relative to the small size of the leaf seedlings, the rapid depletion of soil water content, and, consequently, the increased vulnerability to drought [45,59]. These variations in LT and LA observed here confirm the greater variation in cell size and arrangement and the amounts of structural tissue across the ecosystem. Furthermore, the exposure of the leaf to drought might decrease the Amax, which might shorten the leaves and also the cell elongation due to turgor loss [60,61]. The plants with higher photosynthetic apparatus experienced a higher rate of transpiration and depleted water in the soil at a faster rate than the plants with a relatively lower leaf size. Thus, C gain per unit of leaf area increases at the cost of transpiration rate, ultimately leading to a decline in plant water potential. Moreover, these modifications in leaf size can increase the risk of mortality at a faster rate than plant growth, which suggests that an energetic leaf-level constraint between C acquisition and water depletion was expressed as a trade-off between growth and survival at the plant level [55,59].

Plant leaves are the key organs of photosynthetic CO_2_ assimilation, and leaf area determines light harvesting, which affects the photosynthetic activity [62,63]. At the leaf level, stomatal adjustments allow the plant to perform a suboptimal strategy in which leaves optimize water-use efficiency under rapidly changing environmental conditions and a fixed level of soil potential. As water availability changes, whole-plant C gain and water-use optimization require that the plant reaches a new functional equilibrium in daily patterns of water expenditure in terms of soil water availability [59]. In our study, the stomatal pore length was significantly decreased by drought, as previously reported [64]. Drought may initially inhibit leaf growth and development, significantly decreasing the leaf area [65,66]. Under drought, the plant’s initial response is to exclude the minimum water potential by adjusting its water maintenance between root water uptake and water loss in leaves [67,68]. Our results, however, imply the positive association between SD and LT and suggest that the enhanced LT may produce more guard cells for a given leaf area [65]. Increased LT and the associated increased SD might be useful in enhancing the plasticity to a certain degree under drought [62,65]. These modifications in SL and SD may also be due to the genetic factors and/or plant growth and developments against various environmental variables [64,69]. Loss in stomatal conductance due to reduced SL has been linked to high water conservation and is well established in plants against drought [69].

### 4.3. Effect of Drought on Leaf Hydraulic or Anatomical Traits and Stem Water Status

Our results showed that the variation in LT is described by the variations in spongy and palisade mesophyll VA and adaxial epidermis VA. These variations might be clearly explained by the variations in cell size or in the number of mesophyll cell layers [70,71]. The strong correlation between leaf and mesophyll VA could be correlated to the lower rates of Amax/DM. This may allow the plants to be more competitive during the short growing season [72]. In our study, a reduction in conduit diameter indicates a reduction in water transport efficiency, and such reduction might have been partially counterbalanced by increased conduit density. This would decrease the xylem area, which might directly be connected to the low availability of water [55,73]. Our results suggest that low water transport efficiency requires the construction of a safe xylem to reduce the risk of cavitation, which comes at the cost of substantial C investment in the leaf hydraulic system [74,75,76]. Furthermore, the reduction in VBD might be interpreted as an adaptation to drought, which may be the outcome of the nutrient depletion for the growth of these tissues [77]. In order to maintain leaf growth under drought, plants might alter the leaf hydraulic traits [78]. These alterations in hydraulic traits and transpiration acclimation are a type of functional and anatomical development which might be associated with adaptation to unfavorable environmental conditions [79]. These findings also revealed that the interrelations between the variations in leaf hydraulic traits and drought stress were dependent upon the plant species and the diversity of stress adaptation among the different species. In our study, however, we observed that the variations in hydraulic traits were equally affected by drought across all the studied species. This suggests that there are no clear differences in leaf anatomy between taxa based on sensitivity status, implying that there are other aspects of plant physiology which determine the sensitivity to extreme drought. Thus, there is no strong evidence in our dataset to suggest that the leaf anatomical traits of drought-resistant species are consistently different from those of drought-sensitive species.

In our study, the stem water potentials were significantly altered by severe drought (Figure 4A,B). We observed that the SL was significantly decreased by severe drought, indicating that the species are able to maintain the stem water potential (Ψ_Stem_), regulating the evaporative demand and mitigating the risk of hydraulic failure, which corroborates the previous findings of Attia et al. [80]. In our study, the SL was weakly correlated to Ψ_PD_ and strongly negatively correlated to Ψ_MD_ (Figure 4A,B). The prevention of hydraulic failure by regulating stem water potential is more advantageous under drought than photosynthesis, which holds the risk of hydraulic failure [56,81]. We also observed that all the species had different Ψ_PD_ and Ψ_MD_ across all the drought treatments, with a significant impact of severe drought, but the plants were more dehydrated in midday―suggesting that the pressure drop between Ψ_PD_ and Ψ_MD_ was maintained for water exclusion under drought.

## 5. Conclusions

This study examined the correlations among the plant functional traits known to relate to drought tolerance. A few traits, such as LMF, RMF, and plant biomass, are indicative of the seedlings’ ability to cope with drought conditions. Our findings are that the interrelations between leaf traits and above-ground growth characteristics are more consistent in *F. mandshurica* and *P. koraiensis*. These showed a similar resistant pattern with respect to the above-ground characteristics (i.e., RMF, LMF), suggesting drought tolerance. The species *L. gmelinii*, which exhibited the lowest performance under drought, had the least RMF plus higher LMF, which together resulted in a water-inefficient allocation pattern. However, *T. amurensis*, with higher photosynthetic apparatus under drought conditions, may offer higher evaporative demand relative to the small size of the leaf seedlings, the rapid depletion of soil water content, and, consequently, the increased vulnerability to drought. Furthermore, there is no strong evidence in our dataset to suggest that the interrelations between the leaf morpho-anatomical traits of the drought-resistant species (i.e., *P. koraiensis* and *F. mandshurica*) are consistently different from those of the drought-sensitive species (i.e., *L. gmelinii* and *T. amurensis*). Further studies on adult trees are needed to test the hypothesis that the seedlings of woody species might be more vulnerable to drought conditions relative to those of the adult individuals.

## Figures and Tables

**Figure 1 biology-11-01186-f001:**
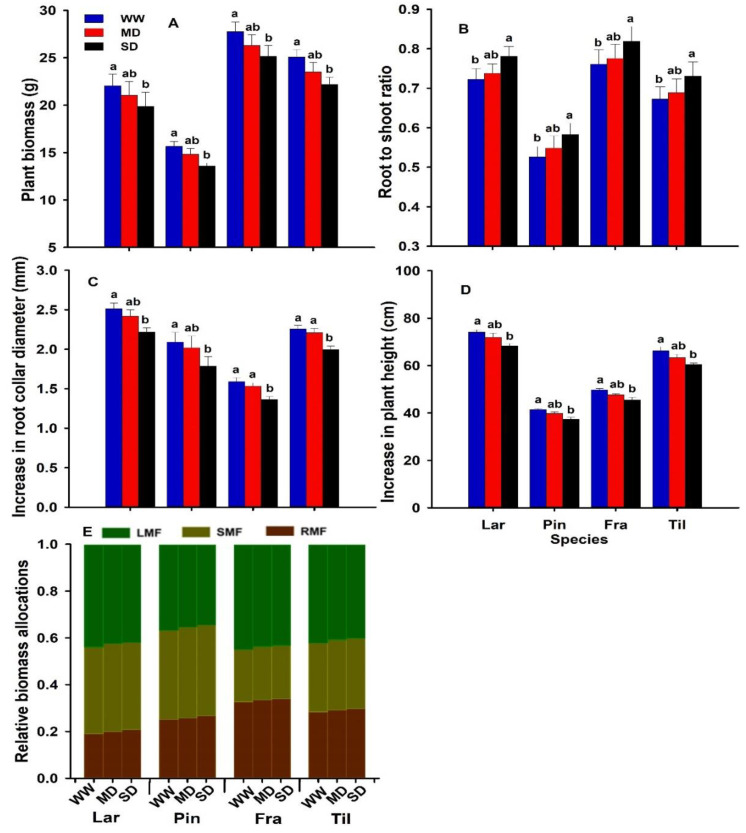
Plant biomass (PB; g) (**A**), root shoot ratio (**B**), increase in root collar diameter (RCD; mm) (**C**), increase in plant height (PH; cm) (**D**), and relative biomass allocation (**E**); root mass fraction (RMF; grey bars), stem mass fraction (SMF; light grey bars), and leaf mass fraction (LMF; green bars) of two-year-old seedlings of *Larix gmelinii* (Lar), *Pinus koraiensis* (Pin), *Fraxinus mandshurica* (Fra), and *Tilia amurensis* (Til) at severe drought, SD, ≤5%; moderate drought, MD, 6–10%; and well-watered, WW, 16–20%.

**Figure 2 biology-11-01186-f002:**
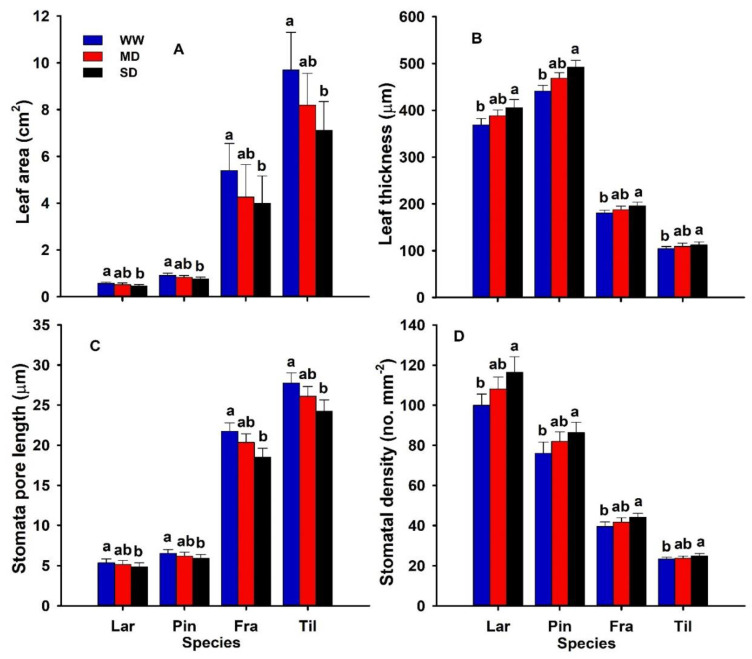
Leaf area (LA; cm^2^) (**A**), leaf thickness (LT; µm) (**B**), stomatal pore length (SL; µm) (**C**), and stomatal density (**D**) (SD; no. mm^−2^) of two-year-old *Larix gmelinii* (Lar), *Pinus koraiensis* (Pin), *Fraxinus mandshurica* (Fra), and *Tilia amurensis* (Til) seedlings at three drought levels (i.e., severe drought, SD, ≤5%; moderate drought, MD, 6–10%; and well-watered, WW, 16–20%). Within species, significant differences between treatments are indicated by different lower-case letters (Tukey’s HSD post hoc; *p* < 0.05; mean ± SE).

**Figure 3 biology-11-01186-f003:**
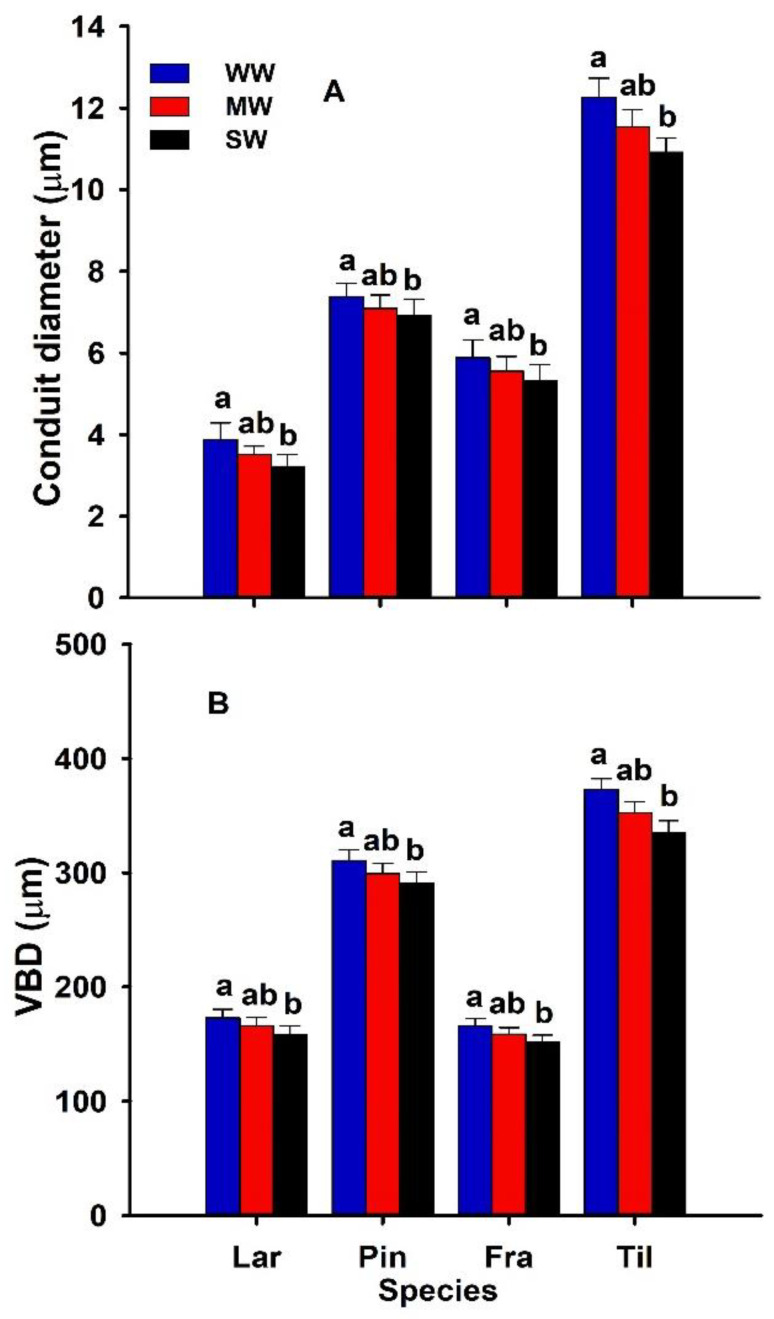
Xylem conduit diameter (XCD; µm) (**A**) and vascular bundle diameter (VBD; µm) (**B**) of two-year-old *Larix gmelinii* (Lar), *Pinus koraiensis* (Pin), *Fraxinus mandshurica* (Fra), and *Tilia amurensis* (Til) seedlings at three drought levels (i.e., severe drought, SD, ≤ 5%; moderate drought, MD, 6–10%; well-watered, WW, 16–20%). Within species, significant differences between treatments are indicated by different lower-case letters (Tukey’s HSD post hoc; *p* < 0.05; mean ± SE).

**Figure 4 biology-11-01186-f004:**
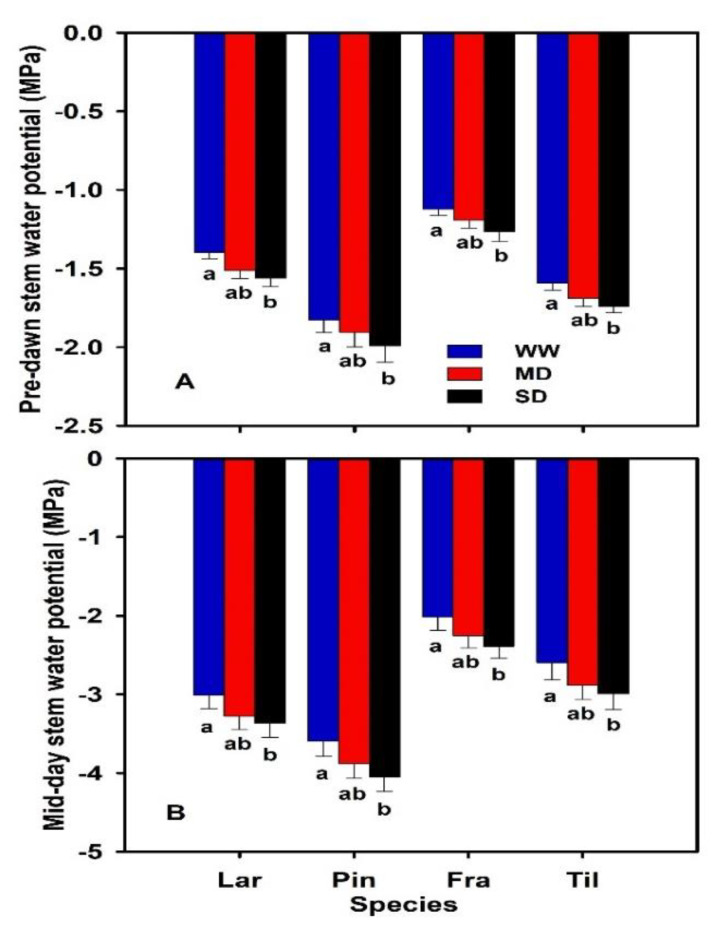
Predawn stem water potential (Ψ_PD_; MPa) (**A**) and midday stem water potential (Ψ_MD_; MPa) (**B**) of two-year-old seedlings of *Larix gmelinii* (Lar), *Pinus koraiensis* (Pin), *Fraxinus mandshurica* (Fra), and *Tilia amurensis* (Til) at three drought levels (i.e., severe drought, SD, ≤ 5%; moderate drought, MD, 6–10%; well-watered, WW, 16–20%). Within species, significant differences between treatments are indicated by different lower-case letters (Tukey’s HSD post hoc; *p* < 0.05; mean ± SE).

**Figure 5 biology-11-01186-f005:**
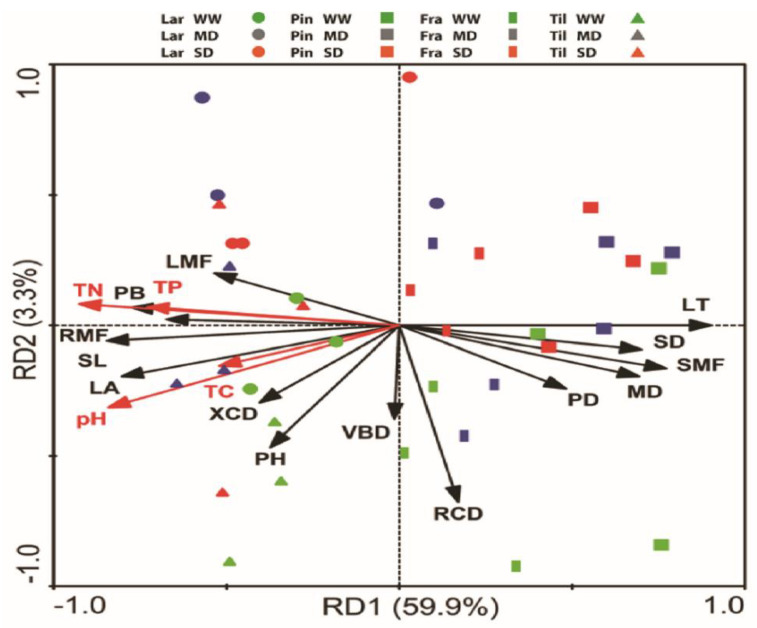
Redundancy analysis (RDA) of morpho-functional plant traits to the soil-related parameters, i.e., soil pH, TN, TC, and TP taken from 0–20 cm soil depth in northeast China. Abbreviations, LT: leaf thickness; LA: leaf area; XCD: xylem conduit diameter; VBD: vascular bundle diameter; SL: stomata pore length; SD: stomatal density; PB: plant biomass; PH: plant height; RCD: root collar diameter; RMF: root mass fraction; SMF: stem mass fraction; LMF: leaf mass fraction; PD: stem water potential at predawn; MD: stem water potential at midday; pH: soil pH; TN: total nitrogen; TC: total carbon; TP: total phosphorus.

**Figure 6 biology-11-01186-f006:**
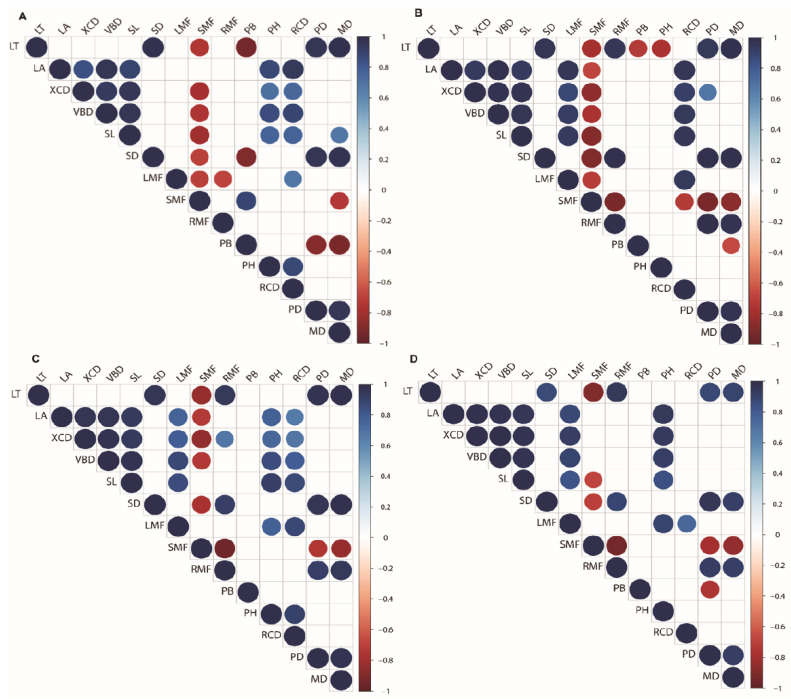
Pearson’s correlation coefficients of morpho-functional plant traits of two-year-old seedlings of *L. gmelinii* (**A**), *P. koraiensis* (**B**), *F. mandshurica* (**C**), and *T. amurensis* (**D**) in northeast China. Abbreviations, LT: leaf thickness; LA: leaf area; XCD: xylem conduit diameter; VBD: vascular bundle diameter; SL: stomatal pore length; SD: stomatal density; PB: plant biomass; PH: plant height; RCD: root collar diameter; RMF: root mass fraction; SMF: stem mass fraction; LMF: leaf mass fraction; PD: stem water potential at predawn; MD: stem water potential at midday.

**Table 1 biology-11-01186-t001:** ANOVA results for tree species, drought intensities, and their interaction effects on leaf morphological and anatomical traits, stem water potentials, and biomass allocation patterns of two-year-old seedlings of *Larix gmelinii* and *Pinus koraiensis* (gymnosperms) and *Fraxinus mandshurica* and *Tilia amurensis* (angiosperms), at three drought levels (i.e., severe drought, SD, ≤5%; moderate drought, MD, 6–10%; well-watered, WW, 16–20%) in NE China. Abbreviations, PB: plant biomass; RCD: root collar diameter; PH: plant height; LMF: leaf mass fraction; SMF: stem mass fraction; RMF: root mass fraction; Ψ_PD_: stem water potential at predawn; Ψ_MD_: stem water potential at midday; LT, leaf thickness; LA: leaf area; XCD: xylem conduit diameter; VBD: vascular bundle diameter; SL: stomata pore length; SD: stomatal density.

Sources of Variation	df	PB	Root: Shoot	RCD	PH	LMF	SMF	RMF	Ψ_PD_
Species (Sp.)	3	**0.001**	**0.001**	**0.001**	**0.001**	**0.001**	**0.001**	**0.001**	**0.001**
Drought (D)	2	**0.010**	**0.045**	**0.001**	**0.001**	**0.033**	0.819	**0.048**	**0.008**
Sp. × D	6	1.000	1.000	0.977	0.999	1.000	1.000	1.000	1.000
Sources of variation	df	Ψ_MD_	LA	LT	XCD	VBD	VBA	SL	SD
Species (Sp.)	3	**0.001**	**0.001**	**0.001**	**0.001**	**0.001**	**0.001**	**0.001**	**0.001**
Drought (D)	2	**0.016**	**0.034**	**0.012**	**0.039**	**0.005**	**0.006**	**0.019**	**0.019**
Sp. × D	6	1.000	0.332	0.668	0.749	0.84	0.347	0.568	0.879

*p* values in the bold indicate significant effects (*p* < 0.05).

**Table 2 biology-11-01186-t002:** Anatomical traits of mature, sun-exposed needles/leaves of two-year-old seedlings of *Larix gmelinii*, *Pinus koraiensis* (gymnosperms), *Fraxinus mandshurica*, and *Tilia amurensis* (angiosperms), at three drought levels i.e. severe drought, SD, ≤5%; moderate drought, MD, 6–10%; well-watered, WW, 16–20% in NE China.

Species	Drought Levels	Palisade-	Spongy-	Abaxial-	Adaxial-
		Mesophyll Thickness		Epidermis	
*F. mandshurica*	WW	73.86 ± 3.2 a	63.35 ± 1.8 a	13.54 ± 1.6 a	8.81 ± 0.5 a
	MD	78.88 ± 3.4 ab	69.61 ± 3.2 ab	16.06 ± 1.2 ab	10.61 ± 1.0 ab
	SD	84.08 ± 5.4 b	72.67 ± 2.6 b	18.58 ± 1.9 b	12.12 ± 1.1 b
*T. amurensis*	WW	50.63 ± 2.4 a	36.41 ± 2.8 a	14.69 ± 1.6 a	8.01 ± 0.6 a
	MD	53.08 ± 2.0 ab	39.87 ± 1.9 ab	17.22 ± 1.6 ab	9.05 ± 0.6 ab
	SD	58.43 ± 1.5 b	43.44 ± 1.8 b	20.37 ± 1.8 b	9.99 ± 0.8 b
*L. gmelinii*	Drought levels	Epi- hypodermis	Mesophyll thickness	Resin duct	
	WW	11.52 ± 1.1 a	22.78 ± 1.2 a	51.35 ± 2.2 a	
	MD	12.73 ± 0.9 ab	24.35 ± 0.8 ab	48.55 ± 1.7 ab	
	SD	13.96 ± 0.8 b	26.11 ± 0.7 b	46.86 ± 1.7 b	
*P. koraiensis*	WW	19.51 ± 1.3 a	23.34 ± 1.3 a	26.42 ± 0.9 a	
	MD	22.30 ± 1.3 ab	24.94 ± 0.8 ab	24.56 ± 1.0 ab	
	SD	24.86 ± 1.2 b	26.47 ± 1.1 b	23.92 ± 0.5 b	

Significant differences between treatments per species and trait are indicated by different lower-case letters (Tukey’s HSD post hoc; *p* < 0.05; mean ± SE).

## Data Availability

The data presented in this study are available on request from the corresponding author.

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
