# Peer review of "Limited Acclimation in Leaf Morphology and Anatomy to Experimental Drought in Temperate Forest Species"

_biology, 2022, doi:10.3390/biology11081186_

Round 1

Reviewer 1 Report

The revised manuscript is much improved compared to the previous version. However, there are some changes that need to be made before it can be considered for publication. These intend mostly to improve the strength of conclusions and thus increase impact.

L 14-15: "tree species (Larix ... amurensis) under three varying ..."; it would also be better to indicate which two are gymnosperms and which two are angiosperms (or just simply write "two gymnosperm and two angiosperm tree species", without mentioning species names.

L 22-31: Much improved. However, you mention species names but not whether they are gymnosperm or angiosperm (see my previous comment - phylum is more important than species) and do not explain what LMF and RMF mean. You might want to consider focusing your abstract (and entire paper?) on differences between gymnosperms and angiosperms.

L 40-41: Is drought not an aspect of climate change?

L 44-46: "economic and ecological losses" is very general. Also, they are not the cause of diversity loss but its outcome.

L 93-95: "in two gymnosperm (species names) and angiosperm (species names) tree species that are dominant in natural and afforested temperate forests in NE China"

L 95-97: In what ways? Are these contrasts related to the gymnosperm-angiosperm difference?

L 101-102: "We studied multiple functional traits" (let the reader be the judge if this is a strong point).

L 108-109: Hypotheses should be more specific. 1) Differences between gymnosperms and angiosperm? Based on what? 2) The hypothesis itself sounds trivial, unless you predict which types/criteria make some traits better predictors than others.

L 184: "see section 2.5"

L 230: post-hoc

L 377-379: Badly written

L 377-396: This whole section is a little hard to follow (too many interrelations and too many variables, multiplied by four species). Try to focus on main trends.

L 544: "However ... traits)" - delete

L 545: "evidenced" - change to "are indicative of"

L 554-556: Rewrite so it does not sound like you make a general conclusion for all tree species based on four species.

Author Response

Reviewer 1

The revised manuscript is much improved compared to the previous version. However, there are some changes that need to be made before it can be considered for publication. These intend mostly to improve the strength of conclusions and thus increase impact.

L 14-15: "tree species (Larix ... amurensis) under three varying ..."; it would also be better to indicate which two are gymnosperms and which two are angiosperms (or just simply write "two gymnosperm and two angiosperm tree species", without mentioning species names.

Thank you. The reviewer’s suggestions have been incorporated i.e. two gymnosperms (Larix gmelinii, and Pinus koraiensis) and two angiosperms (Fraxinus mandshurica and Tilia amurensis) tree species were grown under three varying drought intensities in NE China. 

L 22-31: Much improved. However, you mention species names but not whether they are gymnosperm or angiosperm (see my previous comment - phylum is more important than species) and do not explain what LMF and RMF mean. You might want to consider focusing your abstract (and entire paper?) on differences between gymnosperms and angiosperms.

Thank you for pointing out this. Yes, phylum is more important than species. However, we found contrasting results across gymnosperms and angiosperms. However, based on our results one gymnosperms (Pinus koraiensis) and one angiosperm (Fraxinus mandshurica) performed better than the rest of the two species. Our results are contrasting based on phylum thus we have drafted our manuscript based on species (separately) rather than phylum.

 L 40-41: Is drought not an aspect of climate change?

The sentence has been corrected based on the reviewer’s comment in the revised manuscript.

L 44-46: "economic and ecological losses" is very general. Also, they are not the cause of diversity loss but its outcome.

This sentence has been rewritten.

L 93-95: "in two gymnosperm (species names) and angiosperm (species names) tree species that are dominant in natural and afforested temperate forests in NE China"

Thank you. The sentence has been revised based on the reviewer’s suggestions.

L 95-97: In what ways? Are these contrasts related to the gymnosperm-angiosperm difference?

Our studied species are contrasting drought tolerances based on the functional traits but not based on gymnosperm-angiosperm difference. In our study we observed that P. koraiensis and F. mandshurica are performed better while L. gmelinii and T. amurensis performed poor under drought. We did not observe a clear difference based on gymnosperm-angiosperm differences. Some previous study also found contrasting drought tolerances within gymnosperms. For example, (Ning et al. 2022) reported that L. gmelinii and P. koraiensis are performed differently to water limitation in NE China. Based on their xylem hydraulics P. koraiensis performed better while the growth of L. gmelinii were significantly declined.  

Ning Q-R, Gong X-W, Li M-Y, Hao G-Y (2022) Differences in growth pattern and response to climate warming between Larix olgensis and Pinus koraiensis in Northeast China are related to their distinctions in xylem hydraulics. Agric For Meteorol 312: 108724. doi: 10.1016/j.agrformet.2021.108724.

L 101-102: "We studied multiple functional traits" (let the reader be the judge if this is a strong point).

The sentence has been revised.

L 108-109: Hypotheses should be more specific. 1) Differences between gymnosperms and angiosperm? Based on what? 2) The hypothesis itself sounds trivial, unless you predict which types/criteria make some traits better predictors than others.

L 184: "see section 2.5"

Thanks! Corrected.

L 230: post-hoc

Corrected.

L 377-379: Badly written

Rewritten.

L 377-396: This whole section is a little hard to follow (too many interrelations and too many variables, multiplied by four species). Try to focus on main trends.

Thank you for pointing out this. The whole section has been revised based on reviewer’s suggestions.

L 544: "However ... traits)" – delete

Deleted.

L 545: "evidenced" - change to "are indicative of"

Changed “evidenced” into “are indicative of”

L 554-556: Rewrite so it does not sound like you make a general conclusion for all tree species based on four species.

This sentence has been rewritten.

Reviewer 2 Report

All my comments have been considered. The manuscript has been improved

Author Response

Reviewer 2

All my comments have been considered. The manuscript has been improved.

Thank you for your valuable suggestions and comments for improving the first draft of the manuscript.
